

# Dependency management bots in open-source systems—prevalence and adoption

Linda Erlenhov[1], Francisco Gomes de Oliveira Neto[2] and Philipp Leitner[1]

[1] Department of Computer Science and Engineering, Chalmers University of Technology, Gothenburg, Sweden
[2] Department of Computer Science and Engineering, University of Gothenburg, Gothenburg, Sweden

## ABSTRACT

Bots have become active contributors in maintaining open-source repositories. However, the definitions of bot activity in open-source software vary from a more lenient stance encompassing every non-human contributions *vs* frameworks that cover contributions from tools that have autonomy or human-like traits (*i.e.*, Devbots). Understanding which of those definitions are being used is essential to enable (i) reliable sampling of bots and (ii) fair comparison of their practical impact in, *e.g.*, developers' productivity. This paper reports on an empirical study composed of both quantitative and qualitative analysis of bot activity. By analysing those two bot definitions in an existing dataset of bot commits, we see that only 10 out of 54 listed tools (mainly dependency management) comply with the characteristics of Devbots. Moreover, five of those Devbots have similar patterns of contributions over 93 projects, such as similar proportions of merged pull-requests and days until issues are closed. Our analysis also reveals that most projects (77%) experiment with more than one bot before deciding to adopt or switch between bots. In fact, a thematic analysis of developers' comments in those projects reveal factors driving the discussions about Devbot adoption or removal, such as the impact of the generated noise and the needed adaptation in development practices within the project.

# INTRODUCTION

Bots are becoming prevalent tools in software development environments (*Lebeuf, Storey & Zagalsky, 2018*; *Erlenhov, de Oliveira Neto & Leitner, 2020*), particularly when bots are supportive of costly software maintenance tasks involving, *e.g.*, creating pull requests (PRs) (*Wessel et al., 2020a*), code refactoring (*Wyrich & Bogner, 2019*) or code contributions over time (*Wessel et al., 2018*). Consequently, various studies investigate the impact of adopting a bot in a software development process. Recent work showed that the adoption of bots can have a significant impact on overall project metrics, such as number of PRs created and closed before and after a bot was introduced (*Wessel et al., 2020a*).

Corresponding author
Linda Erlenhov,
linda.erlenhov@chalmers.se

[1] For example, the first edition of the Automated Software Engineering conference was held in 1997.

The underlying challenge such studies face is the difficulty of determining what exactly constitutes a "bot", and to distinguish bots from automation in general (a topic that has been studied in the software engineering community at least since the early 2000s[1]). Two different mindsets appear to be prevalent in existing studies: whereas researchers working on taxonomies or definitions often stress the difference to automation tools, *e.g.*, by requiring bots to have, for example, human-like traits, such as a name (*Erlenhov, de Oliveira Neto & Leitner, 2020*), language (*Lebeuf et al., 2019*), or purpose (*Erlenhov et al., 2019*), more quantitative studies often take a relatively all-encompassing stance where every contribution that is not made directly by a human developer is considered a bot contribution (*Dey et al., 2020b*).

Depending on the study goal, such a wide definition may be exactly what is required. For example, *Dey et al. (2020b)* have proposed an approach to identify bot commits so as to *exclude* them from studies that target human behaviour. For such a study, whether the author of an excluded contribution is a bot or "just" an automation tool is largely irrelevant. However, for work that specifically targets the study of bot contributions and their effect on human developers, it seems central to more clearly delineate between tools that actually exhibit bot-like characteristics (according to existing taxonomies and classification frameworks) and other automation tools that do not.

Therefore, our goal is to (i) investigate some of the approaches above that classify bots and then (ii) verify whether a clearer distinction between bots and automation tools provides insights about the impact of bot activity in a project. Particularly, we leverage widely used impact measures such as PRs and comments to investigate the activity generated by one or more bots in the same project and also the interaction between humans and those bots (*Wessel et al., 2020a*; *Wessel et al., 2018*). Our general hypothesis is that a more refined approach to define and sample bots enables consistent comparison of *one or more bots* in maintaining the *same project* and reveals insights about bot activity (*e.g.*, discussion threads between humans and bots) that are tangential to the expected benefits that any automation tool brings (*e.g.*, creating more PRs and commits).

We investigate that hypothesis in an exploratory empirical study with open-source projects following a multi-method methodology composed of both quantitative and qualitative analysis of bot activity. In order to sample bots we rely on two existing studies that characterise bots: the BIMAN dataset which includes bot commits produced automatically by the BIMAN approach proposed by *Dey et al. (2020a*, *2020b)*, and the bot users' personas introduced in our own earlier work (*Erlenhov, de Oliveira Neto & Leitner, 2020*), which focuses explicitly on how practitioners distinguish bots from automation tools. Below, we summarise our research questions and findings:

- **RQ1 - How much of the dataset includes automation tools that are, according to a more strict definition, not bots?** As a first step, we qualitatively assess a sample of tools from the BIMAN dataset (*Dey et al., 2020a*) through the lens of bot users' personas (*Erlenhov, de Oliveira Neto & Leitner, 2020*). We observe that only 10 of 54 (18.5%) analysed tools would qualify as bots according to our less lenient categorisation

(they would be considered automation tools without human-like characteristics). Further, with one exception, these bots were all dependency management bots.

- **RQ2 - Do similar dependency management bots generate contrasting patterns of activity? Are their pull requests often merged by developers? How often do projects use multiple dependency management bots?** Based on RQ1 results, we further analyse five dependency management bots from the dataset, and mine their activity (created pull requests and corresponding discussion threads) in 93 projects to perform a temporal analysis comparing patterns of bot activity in those projects. We observe that all five analysed bots exhibit similar behavioural patterns. Further, we observe that many projects experiment with multiple dependency management bots and frequently switch between them.

- **RQ3 - What factors guide the discussions about adopting, switching, discarding or using dependency management bots in open-source software?** Based on the temporal analysis from RQ2, we qualitatively investigated a subset of issues and PRs with discussions about the different features and behaviour of the bot, such as usability aspects that conflict with the project's development praxis, or the increase decrease in noise or trust introduced by the bot. Particularly, we map comments about adopting, discarding or replacing a bot to bot traits (*e.g.*, convenience in handling multiple updates) and behaviour (*e.g.*, intrusiveness autonomy to source code changes). Our analysis reveals that open-soure software maintainers are hoping for improved software quality when adopting dependency management bots. Common problems discussed when adopting, using, discarding and switching between these bots are usability issues, such as difficulties to understand or explain how the bot works, or challenges related to noise that overloads the maintainers.

The key contribution of this paper to the state of research is two-fold:

- Firstly, our work shows that there currently is a dissonance between definitions of bots used by different authors and in different study contexts. Our results related to RQ1 indicate that even datasets such as BIMAN, which have explicitly been created to contain "bot contributions", may contain many tools that would not satisfy more strict delineations of what a bot is. This implies that future bot researchers should be explicit about what definition of "bot" they are assuming, and ensure that the dataset they use (or their own data generation method) follows the same definition.

- Secondly, we conduct an empirical investigation (using a combination of quantitative and qualitative methods) on the subset of tools contained in the BIMAN dataset that are indeed classified as bots even following a more strict delineation. We show that these are mostly very similar (dependency management) tools, and provide insights on how and why developers adopt, discard, or switch between such bots.

The remainder of the paper is structured as follows. In "Related Work", we introduce related research on bots in software engineering. In "Study Methodology", we provide a high-level view of our overall methodology, which is followed by a discussion of our main

results relating to the three research questions in "Distinguishing Bots and Automation Tools", "Activity Analysis of Dependency Management Bots" and "What are the Discussed Challenges and Preferenceswhen Adopting, Switching or Discarding Bots?". Based on these results, we summarise and provide a broader discussion of our findings (and their implications for software engineering research) in "Discussion", in which we also discuss the threats to validity. Finally, we conclude the paper in "Conclusions".

## RELATED WORK

Bots are the latest software engineering trend for how to best utilise the scarce resource "developer time" in software projects. However, the term itself is an umbrella term for several different types of tools used in software engineering. In order to classify these tools, several taxonomies have been presented. *Lebeuf et al. (2019)* presented an extensive, faceted taxonomy of software bots. *Erlenhov et al. (2019)* created a more compact taxonomy specifically focusing on bots in software development. A third taxonomy was proposed by *Paikari & van der Hoek (2018)*, with a particular focus on chat bots in software engineering. The different taxonomies offer complementary views to classify and understand bots. For instance, *Paikari & van der Hoek (2018)* targets chatbots, thusm including many facets to classify different types of interaction and direction between the bot and a human. In contrast, *Lebeuf et al. (2019)* defines 27 subfacets covering intrisic, environmental and interaction dimensions to classify bots. Moreover, all those taxonomies are faceted, which allows them to be expanded to accomodate new levels as the field of software bots evolve (*Usman et al., 2017*). Nonetheless, a limitation common to all three taxonomies is that they lack clear, minimal requirements that a tool would need to fulfil to be considered a bot. In a subsequent study, *Erlenhov, de Oliveira Neto & Leitner (2020)* turned the question around and investigated the developers' perception of bots as a concept, and asked what facets needed to be present in order for the developers to look at a tool as a bot. The authors categorised the tools by introducing three personas based on developers' impressions, since there was not one definition that all developers could agree on. These personas each have a set of minimal requirements that needs to be fulfilled in order for them to recognise the tool as a bot-autonomy, chat and smartness. Each persona's bots come with different problems and benefits, and affects the projects and its developers in different ways.

Research in the last years has explored various different dimensions of software engineering where bots may assist developers, including the automated fixing of functional bugs (*Urli et al., 2018*), bug triaging (*Wessel et al., 2019*), creating performance tests (*Okanović et al., 2020*), or source code refactoring (*Wyrich & Bogner, 2019*). This proliferation of bots is slowly creating demand for coordination between bots in a project, which has recently started to receive attention by *Wessel & Steinmacher (2020)* through the design of a "meta-bot".

### Impact of bot adoption

When it comes to adopting tools in the open-source software ecosystem *Lamba et al. (2020)* looked at how the usage of a number of tools spread by tracking badges from the

projects main page. They found that social exposure, competition, and observability affect the adoption. In a recent paper by *Wessel et al. (2021)*, the initial interview study revealed several adoption challenges such as discoverability issues and configuration issues. The study then continues to discuss noise and introduces a theory about how certain behaviours of a bot can be perceived as noise. Even though previous work often speculates that the adoption of bots can be transformative of software projects (*Erlenhov, de Oliveira Neto & Leitner, 2020*), it is still an open research question how exactly bot adoption impacts projects. Previous work from *Wessel et al. (2018)* studied 44 open source projects on GitHub and their bot usage. They clustered bots based on what tasks the bot performed and looked at metrics such as number of commits and comments before and after the introduction of the bots. However, no significant change could be discerned. One reason for this may have been that this study did not sufficiently distinguish between different types of bots, which may be used for very different purposes. Hence, follow-up research (*Wessel et al., 2020a*) focussed foremost on one specific type of bot, namely code coverage bots (1,190 projects out of 1,194), and found significant changes related to the communication amongst developers as well as a in the number of merged and non-merged PRs. This was subsequently investigated further in an interview study (*Wessel et al., 2020b*). These results, that less discussion is taking place, also is what was found by *Cassee, Vasilescu & Serebrenik (2020)* when looking at how continuous integration impacted code reviews. *Peng et al. (2018)* studied how developers worked with Facebook mention bot. The study found that mention bots impact on the project was both positive in saved contributors' effort in identifying proper reviewers but also negative as it created problems with unbalanced workload for some already more active contributors.

## Bot identification

Another area where bot categorisations are directly useful is in the (automated) study of developer activity. Software repository mining studies, such as the work published every year at the MSR conference (https://conf.researchr.org/home/msr-2021), frequently struggle to distinguish between contributions of humans and bots (where the study goal often requires to only include human contributions). Different approaches have recently been proposed to automatically identify bot contributions (*Golzadeh et al., 2021b*; *Dey et al., 2020b*), also leading to the BIMAN dataset, *i.e.*, a large dataset of bot contributions (*Dey et al., 2020a*) which we build upon in our work. One challenge with identifying bot contributions is the presence of "mixed accounts" (*Golzadeh et al., 2021a*), *i.e.*, accounts that are used by humans and bots in parallel. Mixed accounts require an identification of bot contributions on a the individual contribution level (rather than classifying entire accounts). *Cassee et al. (2021)* have shown that existing classification models are not suitable to reliably detect mixed accounts. In general, existing approaches are sufficient if the goal is to identify human contributions. However, as a foundation to study the bot contributions themselves (*e.g.*, to assess bot impact), existing work lacks fidelity, in the sense that they do not distinguish between different types of automation tools and bots, nor between different types of bots.

Our study directly connects to these earlier works. We use the categorisation model proposed in our earlier work (*Erlenhov, de Oliveira Neto & Leitner, 2020*) to further investigate the BIMAN dataset (*Dey et al., 2020a*), particularly with regards to the question of how many of these automated contributions are actually "bots" in a stricter sense of the word. We further quantitatively as well as qualitatively investigate the (dependency management) bots we identified in the BIMAN dataset, further contributing to the discussion related to the impact of bot adoption on open-source projects.

## STUDY METHODOLOGY

To address our study goal, we perform a multi-method study combining different elements. First we perform a qualitative assessment of the BIMAN dataset (*Dey et al., 2020a*) based on criteria for bot classification defined by practitioners (RQ1), followed by a quantitative analysis based on temporal data of the activity of five dependency management bots (RQ2). Lastly, we look closer at specific bot activity within projects by doing a qualitative, thematic analysis of the discussion threads related to bot adoption, discarding and switching. A high-level overview of our methodology can be found in Fig. 1.

We first extract a complete list of unique tools from the BIMAN dataset, which we then rank by usage. The first author of this study then manually categorised the first 70 tools according to our own classification from earlier research (*Erlenhov, de Oliveira Neto & Leitner, 2020*). Only 10 tools are classified as bots. Subsequently, we select five of those bots and sample 50 projects each that used the bot. For these, we use the GitHub API to extract all PRs and issues where the bot was involved (either as issue creator, commenter, or simply being mentioned). This leads to a large database of bot issues and PRs, which we then analyse both quantitatively and qualitatively. Finally, we select a subset of issues that include discussion threads about multiple bots in order to perform a qualitative analysis on the discussion between human contributors of the project.

Since the data of each RQ feeds into the next, more detailed method information is provided directly in "Distinguishing Bots and Automation Tools", "Activity Analysis of Dependency Management Bots" and "What are the Discussed Challenges and Preferenceswhen Adopting, Switching or Discarding Bots?", such as the choice of dependency management bots and filtering of issues in our datasets. The data collected, and scripts used for analysis can be found in our replication package (*Erlenhov, de Oliveira Neto & Leitner, 2021*). (https://doi.org/10.5281/zenodo.5567370).

## DISTINGUISHING BOTS AND AUTOMATION TOOLS

We now discuss our first research question, an analysis of whether the existing BIMAN dataset of bots aligns with the bot characteristics listed by practitioners in our previous work. Specifically, we are interested how much of the dataset includes pure automation tools.

### Data collection

We started from the BIMAN dataset which includes over 13 million commits from 461 authors. We then extracted the authors and sorted them by the number of GitHub

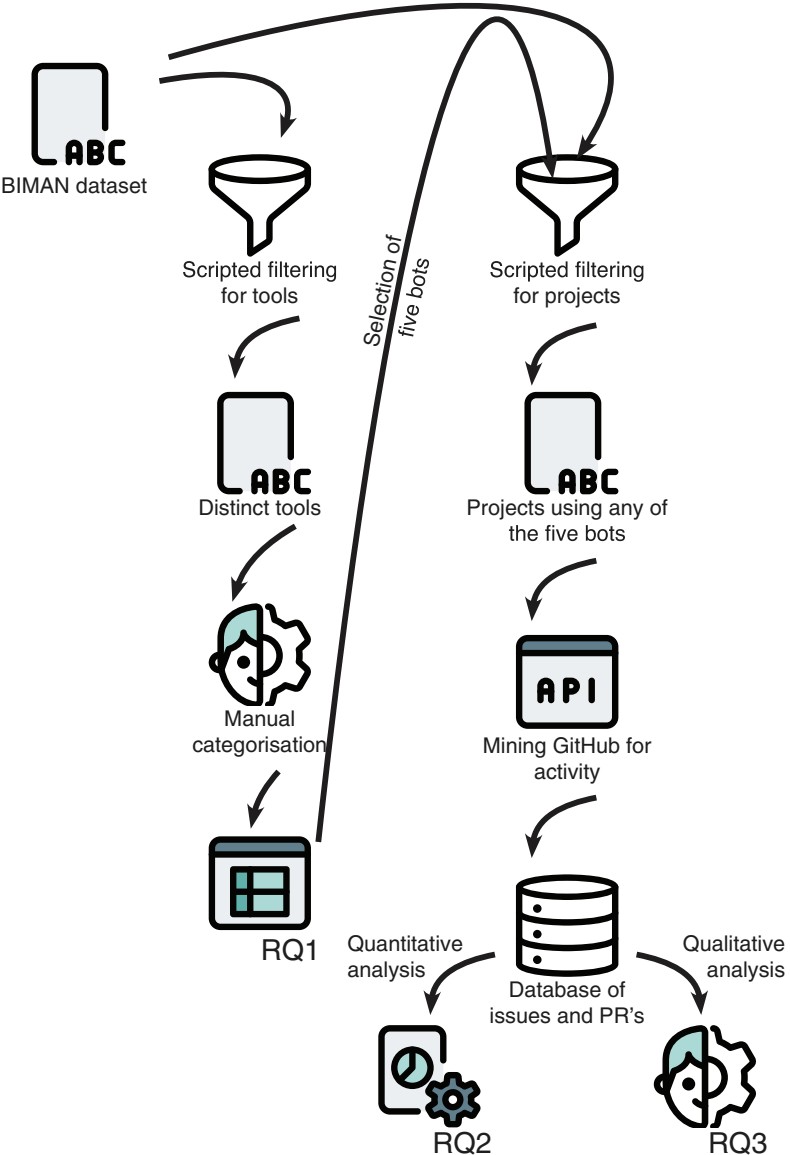

**Figure 1 Overview of our methodology, including the different sources for data collection and their connection to each research question.** Icons made by Pixel perfect from www.flaticon.com.

organisations adopting each tool as a proxy of popularity or importance. However, initial analysis showed that the dataset contained duplicate tools (the same tool acting under multiple identities). We resorted to manually merging identities of the first 70 tools in the ordered list, which after merging, produced a final table consisting of 54 unique tools associated with 89 different authors.

## Analysis and interpretation approach

We analysed these 54 tools manually using the flow-chart to characterise bots proposed in our previous work where we conducted an interview study and a survey with practitioners (*Erlenhov, de Oliveira Neto & Leitner, 2020*). The flow-chart contains five

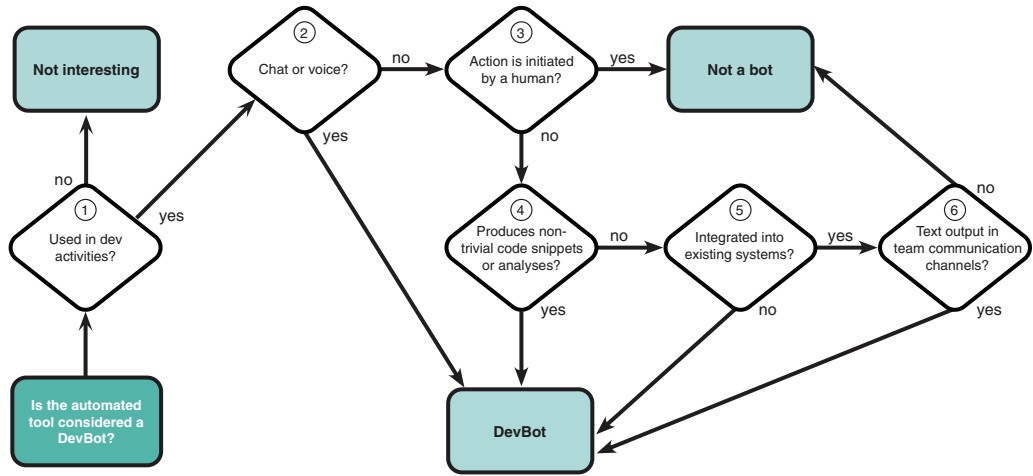

**Figure 2 Decision flow-chart.** Adapted from *Erlenhov, de Oliveira Neto & Leitner (2020)*.

decision blocks with the goal of deciding if the tool would be considered a bot by any of the three personas modelled in the study: Charlie (a bot communicates *via* voice or chat), Sam (a bot does something "smart"), and Alex (a bot works autonomously). Furthermore, the classification implicitly assumed that bots would need to be used for a software engineering task.

For our categorisation, we adapted this decision model slightly (see Fig. 2). We added a decision to first check if the tool was actually used for a software engineering task. Further, since the goal of our study is to decide if a tool is a bot or an automation tool, we were less interested in the specific persona and classified all types of bots simply as "DevBots" with no further distinction.

As the BIMAN dataset only contains commit data, we resorted to manually query additional information (GitHub user profiles, documentation, the tool's external website, developer comments, *etc.*) to arrive at a classification decision for each tool. Examples of additional information used in the classification can be found in Figs. 3 (GitHub) and 4 (tool's external website).

## Results

Following the flow-chart we began by investigating whether the tool was actually used in a software development related task ((**1**) in Fig. 2). Not all tools passed this check—an example of a tool from the dataset that failed this criterion is fs-lms-test-bot. The tool updates repositories with a .learn-file (https://learn.co/lessons/standard-files-in-all-curriculum-lessons) that contains metadata about the project and is added so that participants at a bootcamp style coding school can easily identify what type of repository they are looking at.

Step (**2**) asks if a tool uses chat or voice. For most tools, this proved difficult to determine, and even for promising candidates (*e.g.*, the JHipster bot (https://github.com/jhipster/jhipster-bot)) we found that the part of the tool that produced the git commits that

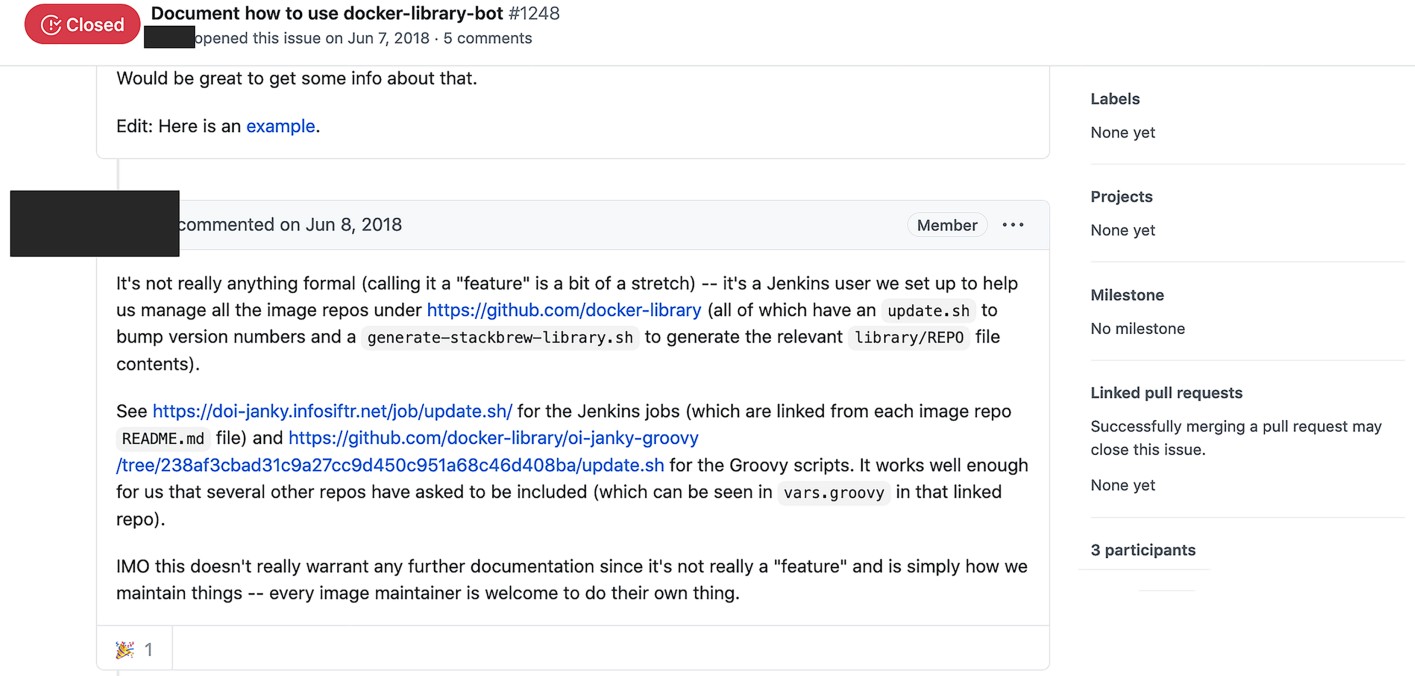

**Figure 3 Example of a GitHub source used to classify the docker-library-bot tool.** (https://github.com/docker-library/docs/issues/1248). The screen shows an issue explaining what the tool does.

we were observing was unrelated to the chat bot. We concluded that, given our analysis data (git commits), this check is not of high value.

Step **(3)** asks if the automated tool initiated by humans. One tool that was considered as automation tool rather than bot because of this check was the Bors bot (https://bors.tech/), which (despite its name), only becomes active when explicitly triggered by a human developer.

In step **(4)**, we investigated if the tool produces nontrivial code snippets or analysis? While clearly a judgement call, we did not consider the output of any tool in our sample to be sufficiently complex or "smart" in the spirit of the original classification model.

Step **(5)** asks if the tool is integrated into existing systems. Examples of tools that failed this check is one of the numerous build helpers, whose only task is to update the code with release versions when someone explicitly initiates this (https://github.com/docker-library/docs/issues/1248).

Finally, the last check in step **(6)** asks if the tool creates text output in team communication channels. Similar to step 2, this proved difficult to determine, as we did not have access to relevant team communication channels. One tool that did emerge as a bot after this check is the Whitesource bot (https://github.com/apps/whitesource-bolt-for-github), which creates one initial commit and after that communicates *via* issues.

On the final list of 54 tools, only 10 tools were (clearly) judged as bots according to the persona-oriented classification model. Table 1 lists these bots and a sample of tools that

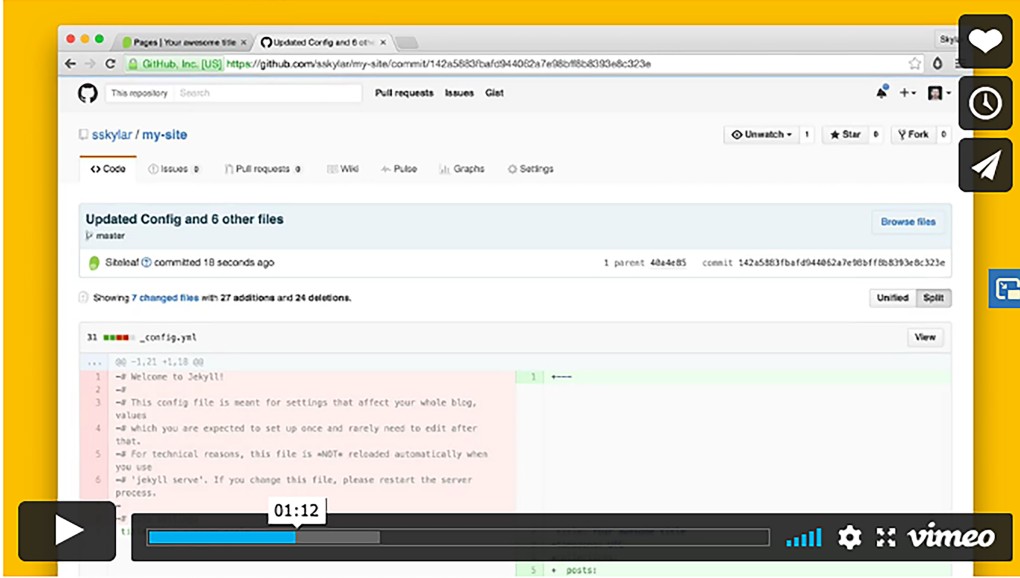

Watch Connecting GitHub and Siteleaf on Vimeo

This tutorial will show you how to connect and sync an existing Jekyll site from GitHub to Siteleaf, so you can edit content and preview your site in the cloud.

If you are new to Jekyll, you may want to start with our Jekyll from Scratch first tutorial to catch up on the basics.

## What is GitHub sync?

When developing your site, you'll generally want to keep your theme and content in sync so you can see how everything looks in context.

**Figure 4** **Example of an external source used to classify the tools.** (https://www.siteleaf.com/blog/connecting-github/). The screen shown is the investigated tool's webpage with a video that implicitly describes how the commits to GitHub are created.

**Table 1 Identified bots and a sample of tools evaluated as automation tools.** Numbers refer to checkpoints in the flow-chart in Fig. 2. Question marks represent checkpoints that we could not answer due to limited information about the corresponding tool.

| Name | (1) | (2) | (3) | (4) | (5) | (6) | Evaluation |
|---|---|---|---|---|---|---|---|
| Whitesource-bot-for-Github | Yes | No | No | No | No | Yes | Bot |
| Greenkeeper | Yes | No | No | No | Yes | – | Bot |
| Dependabot | Yes | No | No | No | Yes | – | Bot |
| Renovate bot | Yes | No | No | No | Yes | – | Bot |
| Pyup bot | Yes | No | No | No | Yes | – | Bot |
| imgbot | Yes | No | No | No | Yes | – | Bot |
| DPE bot | Yes | No | No | No | Yes | – | Bot |
| Snyk bot | Yes | No | No | No | Yes | – | Bot |
| Depfu | Yes | No | No | No | Yes | – | Bot |
| Scala Steward | Yes | No | No | No | Yes | – | Bot |
| fs-lms-test-bot | No | – | – | – | – | – | Not related |
| Bors | Yes | ? | Yes | – | – | – | Automation |
| docker-library-bot | Yes | No | No | No | Yes | – | Automation |
| Siteleaf | Yes | No | No | No | No | No | Automation |
| JHipster bot | Yes | ? | – | – | – | – | Undetermined |

were judged as automation tools. We conclude the following from this classification exercise:

- Only a small fraction (10% of 54%, or 18.5%) of analysed tools clearly qualify as "bots" according to a stricter definition. A large majority are, often fairly conservative, automation tools that have been re-branded as bots, and exhibit little qualitative difference to the kinds of scripts that developers have used for a long time as part of their development, build, and deployment processes.
- Interestingly, this includes many tools that are explicitly called "bots" as part of their names, *e.g.*, the Bors bot or docker-library-bot. Hence, researchers that are interested in investigating bots in a stricter sense should not rely on tool names as primary way to identify bots.
- It is evident that the tools that we actually classified as Devbots (*e.g.*, dependabot, renovate, or greenkeeper) are very similar. More specifically, nine out of these ten bots are dependency management bots on some form. In one case—Snyk and Greenkeeper—one bot was acquired by the other in 2020 (https://snyk.io/blog/snyk-partners-with-greenkeeper-to-help-developers-proactively-maintain-dependency-health/).

## ACTIVITY ANALYSIS OF DEPENDENCY MANAGEMENT BOTS

Based on these findings, we now turn towards a more qualitative investigation of the (dependency management) bots we have identified (RQ2).

## Data collection

We collected data on a subset of the bots identified in "Introduction". Specifically, we selected *Dependabot*, *Greenkeeper*, *Renovate*, *Depfu*, and *Pyup* for deeper quantitative analysis. For each bot, we first compiled a list of all projects in the BIMAN dataset (*Dey et al., 2020a*) that had at least one commit by the selected bot. We sorted these project lists by GitHub watchers, and the first author manually sampled the highest ranked 50 projects for each bot that matched four inclusion criteria. First, the project needed to be a project with actual source code and not a data repository. An example of an excluded project is the `remoteintech/remote-jobs` project which is a list of companies that support remote work. Second, each project had to have more than one issue or PR related to the bot when searching in the issues PR tab on GitHub. Third, the project had to not already been included under another name. Examples of those projects are `kadirahq/paper-ui`, `storybooks/react-storybook` and `storybookjs/storybook`, which took up three positions in the ranked list, but they all point to the same project. Lastly, the project's main language had to be English since the comments from selected projects are used for our qualitative analysis in RQ3.

We observed that the resulting lists of bot-using projects were overlapping, leading to 232 unique projects (from a theoretical maximum of 5 * 50 projects). We consequently downloaded all issue and PR data since the launch of the project until 2021-03-31 for all issues where at least one of our bots was mentioned in the issue text or comments, *or* where at least one of the bots was the author of at least one issue or comment. We downloaded (i) all issue information, (ii) all comments on these issues, and (iii) all merge events related to these issues *via* the GitHub REST API, and stored the resulting JSON data in a MongoDB database for latter processing and analysis. In a last round of filtering we removed all projects that had fewer than 100 issues or PRs, resulting in 93 unique projects. It should be noted that, even though we specifically selected 50 projects for each bot, concrete projects often used a multitude of the study subject bots at different points in the project lifetime.

Table 2 summarises our sample of bot activity in terms of the number of issues PRs and comments created by bots or human contributors, as well as the time period comprising the data. In other words, we refer to bot activity as any issue, PR or comment where one of the selected bot was either the author or was mentioned.

## Analysis and interpretation approach

In order to compare the activity of different bots, we analyse the issues or PRs authored by those bots in the selected projects over the years. This allows us to see increasing decreasing trends of bots usage. Additionally, we analyse how human contributors react to this activity by verifying the proportion of merged PRs that were created by bots and a survival analysis of the issues created by bots. A survival analysis is often used in biology to investigate the expected duration of time until an event occurs (*Kaplan & Meier, 1958*) and, has been used in similar types of analysis in software engineering (*Lin, Robles & Serebrenik, 2017*; *Samoladas, Angelis & Stamelos, 2010*). Our survival analysis measures the

**Table 2 Number of issues, comments and projects for each bot.** There are 93 unique projects in our dataset, but many projects have used multiple bots at some point.

| Issue Author | Projects | Issues | Comments | Period | Years |
|---|---|---|---|---|---|
| Dependabot | 76 | 21,345 | 13,763 | 2017–2021 | 4 |
| Depfu | 16 | 1,346 | 1,032 | 2017–2021 | 4 |
| Greenkeeper | 34 | 3,015 | 2,273 | 2015–2020 | 5 |
| Human | 76 | 1,168 | 30,481 | 2013–2021 | 8 |
| Pyup | 22 | 3,075 | 1,690 | 2016–2021 | 5 |
| Renovatebot | 39 | 12,209 | 2,825 | 2017–2021 | 4 |
| Total | 93 | 42,158 | 52,064 | – | – |

number of days until an issue is closed. We compare the expected duration of PRs created by bots and those created by humans.

Lastly, we analyse overlapping bot activity by comparing (i) projects using multiple bots, as well as (ii) how the bot activity overlap over time. Particularly, we filter projects in which one or more issues were created by two or more bots over the period of, at least, 1 month.

## Results

Figure 5 shows the number of issues and PRs created by each bot over the years. Depfu, Greenkeeper and Pyup have a similar trend beginning with an increase in usage and following a slow decrease in its usage. In parallel, both dependabot and renovatebot have an increasing trend in activity. Most of the issues in our dataset were created by dependabot or renovatebot, indicating a prevalence of such bots among the 93 projects in our dataset.

Figure 6 shows the proportion of merged PRs created by each author. Note that roughly half of the PRs created by humans were merged into the projects. This is surprising as literature reports that PRs created by bots are less likely to be merged than those created humans, whereas here they are the same (*Wyrich et al., 2021*). However, recall that our data collection strategy entailed downloading only issues where bots were involved in some way. Hence, even the human-created issues are not necessarily representative of all issues, as they have still been sampled as issues that somehow involve bot activity (even if not as issue creator). Renovatebot was the only author in which most of the PRs were actually merged (76%), whereas depfu had the lowest percentage of merged PRs (17%).

We also compare the status of the issues created by different bots or humans to check whether there are differences in how long it takes to close those issues. Figure 7 shows a survival curve of the created issues. A survival curve reveals the probability $p(S)$ that an event $S$ occurs (*i.e.*, closing an issue) over a period of time. For consistency, we only consider issues that: (i) lasted at least 1 day, hence avoiding issues closed shortly after creation (*e.g.*, auto-merge dependency updates), (ii) issues created before the date 2021-03-31, or (iii) closed under 120 days in our dataset.

We use a Kaplan–Meier (KM) curve which is a non-parametric statistics to estimate the survival function based on the time period until an event occurs (*Kaplan & Meier, 1958*).

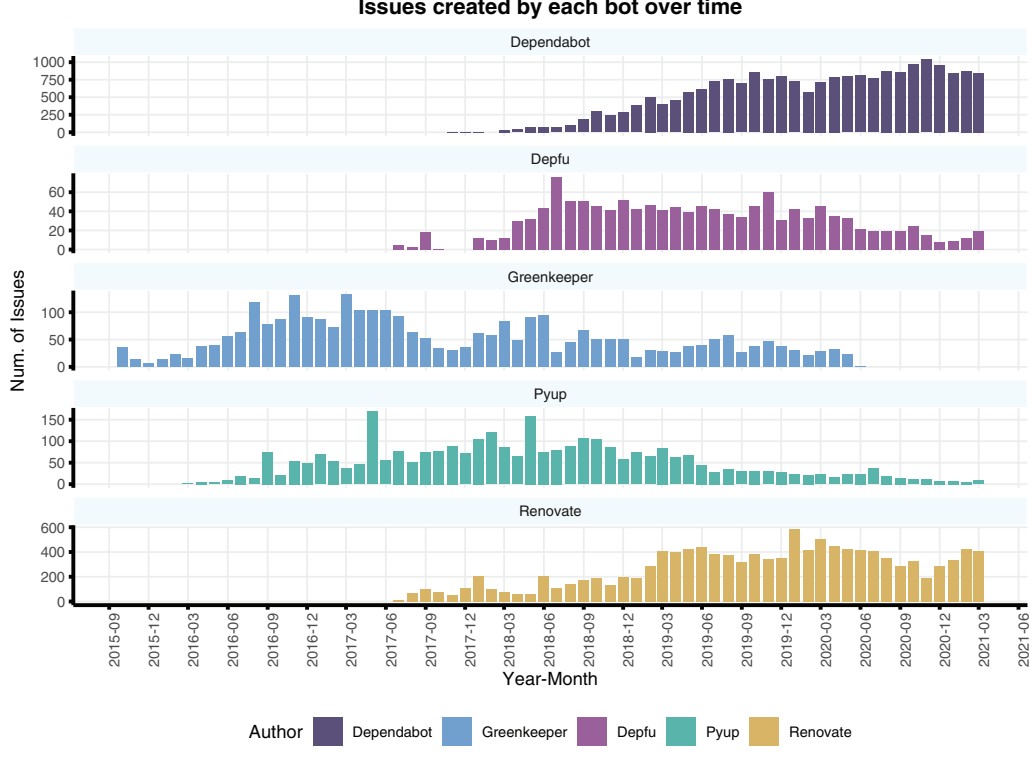

**Figure 5 Number of issues or PR created by each bot throughout the years in our dataset.** Note that the y-axis have different scales to make it easier to compare trends per bot.

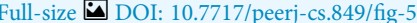

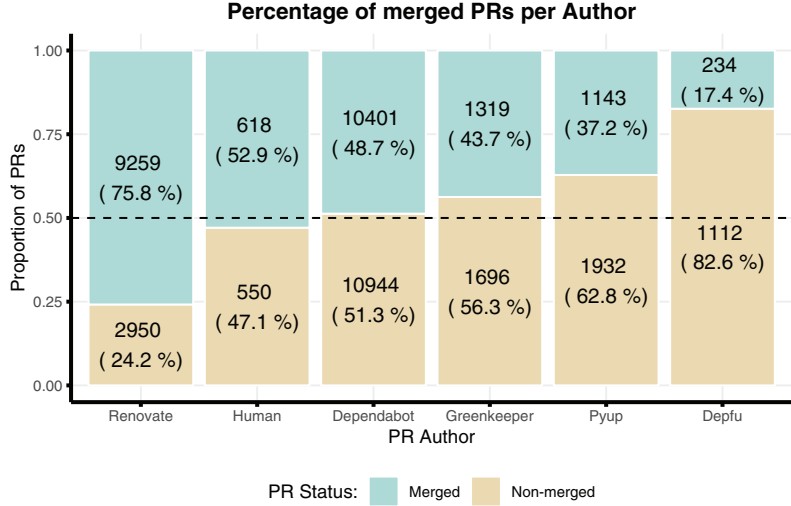

**Figure 6 Proportion of merged PRs created by each author in our dataset.**

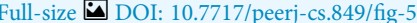

One of the advantages of the KM is to adjust the estimations for *censored events*, which occur when information about the analysed subject is unknown, due to, *e.g.*, missing information about the subject in the dataset. In our case, censored events are issues that

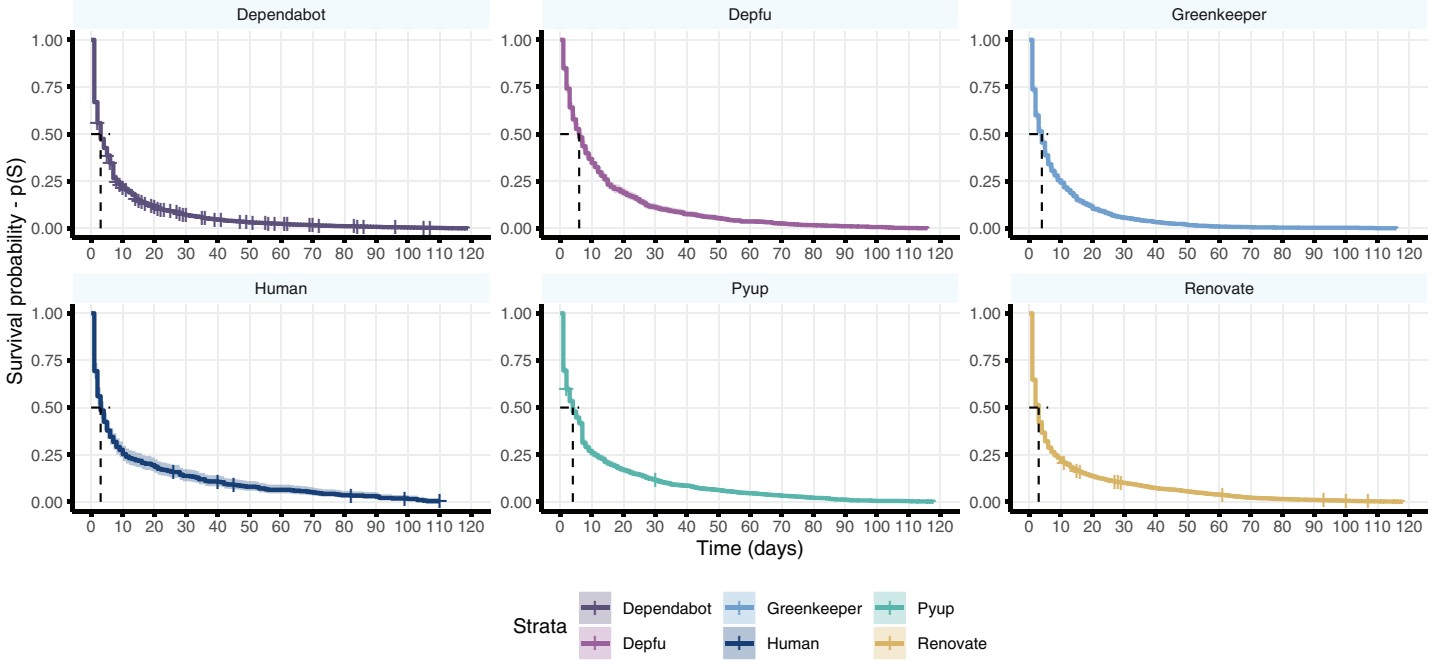

**Figure 7 A survival curve for issues created by different authors in our dataset.** The curve indicates the probability (*y*-axis) of an issue being closed after a number of days (*x*-axis). The ticks in the curve represent censored events, which are issues that were not closed until our limit date (March 31st, 2021). The dashed line shows the median (*p* = 0.5) number of days until an issue is closed.

[2] Left-censored events are those in which data about the first instance of the event, *e.g.*, creation of an issue, is missing. We have no left-censored events in our dataset.

remain open after our limit date (*i.e.*, right-censored)[2]. For instance, we consider censored events those issues that were not closed but were created before within 120 days before our limit date (*i.e.*, our dataset does not include information on whether the issue was indeed closed).

For all issue authors, we see the same pattern in which the issues are most likely to be closed within 5–6 days from the date in which they are created. Dependabot and Renovatebot have more censored events in our dataset because they are also the bots with more recent activity, such that a large number of issues were opened around our limit date. Particularly, there is not a clear difference in the number of days in which bot or human created issues are closed.

Another interesting question our data can answer is to what extent projects use multiple dependency management bots in an overlapping manner (*i.e.*, at the same time). Intuitively, since the basic functionality of the bots is very similar, this should not be a common occurrence. However, when projects switch between bots, a certain overlap may occur.

Table 3 shows the number of projects that use one or more bots, along with the number of months with overlapping bot activity. It is interesting to observe that most of the projects used two of our investigated bots (58%) even though the number of months in which the bots actually work in parallel, for those projects, is expectedly small (13%—242/1783). In other words, the projects used 2 bots at the same for 13% of their

**Table 3 Number of projects that use one or more bots.** For each row, we add the number of months in which more than one bot authored an issue or PR in the project. We also present mean, median and standard deviation (SD) for overlapping months per project.

| Bots used | Projects | Number of months | | Summary on overlapping months | | |
| | | No overlap | Overlap | Mean | Median | SD |
| --- | --- | --- | --- | --- | --- | --- |
| 1 | 21 | 610 | – | 0.0 | 0.0 | 0.0 |
| 2 | 54 | 1,541 | 242 | 4.5 | 2.0 | 8.6 |
| 3 | 14 | 421 | 81 | 5.8 | 2.5 | 6.3 |
| 4 | 4 | 93 | 120 | 30.0 | 16.5 | 33.7 |

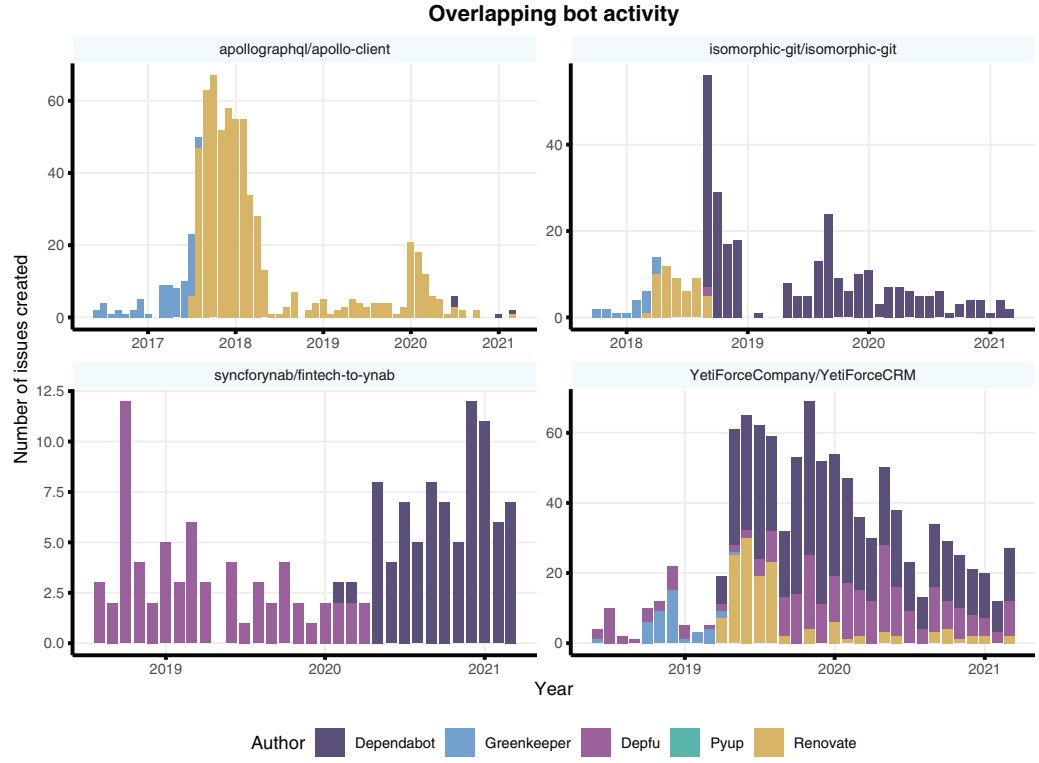

**Figure 8 Sample of projects with a variety of overlapping bot activity.** Different projects have used bots in parallel (*e.g.*, YetiForceCompany), or switched among different bots over the years.

months. In contrast, the few projects that use four bots are using two or more bots in parallel a majority of the time (56%—120/213). We did not see any project that used all of the 5 investigated bots.

The descriptive statistics in Table 3 reveal high variance per project, such that there is great disparity between mean and median. In other words, the overlapping activity varies per project and follow contrasting patterns. We selected a few projects with varied patterns of overlapping bot activity and present them in Fig. 8.

Three of the selected projects indicate that the overlap is specific to transition months. This pattern suggests that developers try out different bots months prior to switching between them (see `apollographql/apollo-client`, `isomorphic-git/isomorphic-git` and `syncforynab/fintech-to-ynab`). Another pattern is a multitude of parallel bot activity, as shown by `YetiForceCompany/YetiForceCRM` in which 2 or 3 bots are constantly being used in parallel throughout years of development.

Since we do not have access to interview the projects' developers, we cannot analyse the factors behind those different patterns. Nonetheless, the patterns reveal a risk when choosing specific dates and counting month intervals before and after bot contributions. The risk is that static timeframes can hide team learning effects from trying out similar bots before the chose timeframe, or miss on confounding effects of multiple bots being used in parallel within a static time frame.

Our analysis of RQ2 verifies the proportion of activity from different dependency management bots, as well as how this activity is consumed by humans by, *e.g.*, merging PRs or closing issues created by the bots. Overall, we did not detect major contrasting patterns or preferences between the investigated bots. That is, we have observed that the usage and contribution patterns of the investigated dependency management bots were largely similar. The main differences were that: (i) Dependabot and Renovate are more popular than the other bots and are increasingly being used by many projects, and (ii) Renovate has more merged issues (75% of merged PRs), whereas Depfu has the least number of merged PRS (17%). Moreover, our survival analysis reveals that most issues are closed before 5 days for all the analysed issues, including those authored by humans in which bots were involved or mentioned in the discussion thread.

Lastly, most projects use 2 or more bots with overlapping activity. However, this overlap varies across projects and indicates different patterns of usage. Based on the findings above, we subset those overlapping months in order to analyse discussion threads and identify which factors drive the developers' decision to adopt, remove or switch between bots.

## WHAT ARE THE DISCUSSED CHALLENGES AND PREFERENCES WHEN ADOPTING, SWITCHING OR DISCARDING BOTS?

Here we investigate the third research question: What factors guide the discussions about adopting, switching, discarding or using dependency management bots in open-source software? Throughout the section we refer to issue identifiers and corresponding URLs specified in Table A1 in the Appendix.

### Data collection

From the dataset used for RQ2, we used the following method to select issues and PRs for our qualitative analysis. First, we identified all issues and PRs that: (i) had two or more bots being mentioned in the comments, or (ii) were created by humans and mention more than one bot in the issue body. In order to include discussion threads about usage of single bots, we manually selected *circa* 30 issues (six issues per investigated bot). A manual inspection of the issues allowed us to include the discussion threads about the

usage of the bot, and remove those about project-specific dependency updates. An example of an PR that was not included is a Dynamoid PR were the discussion is on enabling others to use the bot to update the Dynamoid package dependency in their projects by changing something in the Dynamoid project [Dyn-215].

We further performed snowballing to include issues outside our project sample (*e.g.*, comments such as "see discussion here" that were linked to other issues). Ultimately, the dataset for RQ3 included 109 issues and PRs (included in the replication package (*Erlenhov, de Oliveira Neto & Leitner, 2021*)). (https://doi.org/10.5281/zenodo.5567370). The issues had a mean of 9 (median 7) comments. In total, our analysis is composed of 181 codes extracted from those issues and PRs.

## Analysis and interpretation approach

For our theme analysis, we started by capturing the type of conversation that took place in the each issue. We used four *conversation labels*: adopt, use, switch or discard. The most common case was that one issue contained one conversation, but in some cases we found that a single issue contained multiple logical conversations. For example, an issue in `HypothesisWork/hypothesis` [Hyp-747] started as a conversation about the *usage* of a bot, but later became a conversation about *switching* bots after the developer of another bot decided to join the conversation.

In parallel to identifying conversation labels, we performed open and axial coding where we divided the conversations up into excerpts of relevant information (codes) and assigned a second category of code labels named *content labels* to build our thematic map. Open coding allowed us to generate and vary the categories to classify the codes, whereas axial coding enables sorting of the coded data in new ways by identifying relationships between those categories (*e.g.*, themes and sub-themes) (*Stol, Ralph & Fitzgerald, 2016*). Consequently, our list of code labels was not fixed in the beginning and changed as we reviewed more discussion threads in our dataset.

We based our initial content labels on the bot-related benefits and challenges identified in our earlier work (*Erlenhov, de Oliveira Neto & Leitner, 2020*). Then, we iteratively switched between axial and open coding as new sub-themes were identified. In order to agree on a set of code labels, the first and second authors discussed and coded together roughly 10% of the comments in the dataset. Then, the first author coded the remainder of the dataset. However, due to the open and axial coding, new content themes would surface, hence, triggering another round of discussions between the first and second authors to reach a new agreement on the new set of code labels. This process continued until we reached theory saturation, *i.e.*, no new code labels were created as we sorted codes into the categories. The final table of content code labels and corresponding themes is presented in Table 4.

In summary, we extracted a number of excerpts each assigned with two code labels— one for the *conversation* to keep the context of the discussion thread and a second label to capture the *content* of the excerpt. These excerpts were then sorted into *themes by content*. Each codes and their corresponding conversation and content labels are shared in our replication package.

**Table 4 Description of each code label used in our qualitative study.** For each content sub-theme, we also include the number of codes observed in our dataset.

| Theme | Sub-theme | Codes | Description of Comments or Issues |
|---|---|---|---|
| Promote bot | Creator input | 13 | Bot creator joins the discussion thread to clarify information about their bots. |
| | Company/Project credibility | 9 | Comments regarding whether the bot was developed or sponsored by a reputable company or project. |
| Usability | Setup and configuration | 10 | Technical discussions about introducing and maintaining the bot in the project. |
| | Unistall | 7 | Technical discussions about removing the bot and its artefacts from the project. |
| | Understanding features | 14 | Comments regarding the comprehensibility of features offered by the bot. |
| | Clashes in ways of working | 21 | Discussions about changes in the development process caused by the bot. |
| | Bugs | 4 | Comments regarding faults and failures caused by the usage of the bot. |
| Noise | Annoyance | 5 | Discussions that mention whether the notifications created by the bots are disruptive. |
| | Countermeasures | 10 | Comments suggesting fixes to reduce the notifications created by the bot. |
| | Additional work (for resources) | 5 | Discussion about increased workload on project resources caused by the bot (*e.g.*, build time, tests). |
| | Additional work (for people) | 12 | Discussions about increased workload on humans maintaining the project caused by the bot. |
| Benefits | Improve quality | 10 | Comments about the functional and non-functional improvements caused by the bot. |
| | Handling tasks at scale | 2 | Discussion about enabling development tasks to be performed at higher scales |
| | Automation of tedious tasks | 1 | Comments regarding the bots automating manual and laborious tasks done by developers. |
| | Information retrieval | 2 | Discussion about improved accessibility and availability of project information. |
| Trust | Trustworthy | 7 | Conveys confidence on the bot's agency. |
| | Non-trustworthy | 9 | Conveys unease or suspicion about the bot's agency. |
| Features | Supported features | 19 | Describes features offered by the bot. |
| | Missing features | 21 | Describes features not offered by the bot. |

## Results

A summary of our themes (content labels) and their relation to the conversation labels is shown in Table 5. Our results showed that from the benefits described by *Erlenhov, de Oliveira Neto & Leitner (2020)*, *improved quality* was the main driver for (dependency) bot adoption (primarily related to security and bugfixes). We also expected to find cases related to support *handling tasks at scale*, since adopting a dependency management bot should in principle also allow projects to handle dependency upgrades more easily. Instead, we found that in many cases projects experienced an increase in the load put on maintainers and resources, especially since our studied bots also introduce significant noise in the form of additional work due to numerous PRs.

*"The main driver for this change is to reduce maintenance burden on maintainers, and I really appreciate the effort. However, [redacted]'s comment made me realise that it might have the opposite effect."* -[Dja-2872]

The noise theme was the single theme associated with most coded excerpts related to stop using a bot (*i.e.*, discard). Following our results, the number of PRs generated by the bot is in itself unproblematic, but the bot is perceived to add noise when too many PRs are perceived as irrelevant. However, in some cases the project just accepted that this is just

**Table 5 List of themes and the corresponding number of codes (comments excerpt) associated to each theme and conversation labels.**

| Themes | Adopt | Use | Switch | Discard | Total |
|---|---|---|---|---|---|
| Promote bot | 12 | 3 | 7 | 0 | 22 |
| Usability | 17 | 22 | 14 | 3 | 56 |
| Noise | 12 | 9 | 5 | 6 | 32 |
| Feature | 15 | 6 | 18 | 1 | 40 |
| Benefits | 15 | 0 | 0 | 0 | 15 |
| Trust | 8 | 5 | 2 | 1 | 16 |
| Total | 79 | 45 | 46 | 11 | 181 |

"how bots work". In one case, the developer considered the dependency management bot more as a source of information on existing outdated dependencies than actually trusting it to actually update them [Rea-2673-1]. However, in several cases, the initial load produced by the bot was so large that the projects kept postponing the initial PR for several months—at which point the PR was considered outdated, and the project decided to just discard the bot and start over with a new one [Str-2433].

Our study also reveals multiple *countermeasures to overcome bot noise*, such as (i) limiting the number of simultaneously open PRs from the same bot, (ii) batching the PRs in a smart way, or (iii) letting the bot auto-merge PRs when certain criteria are fulfilled. Evidently, the first and the second approach require developers to decide which PRs the bot was supposed to open (or how to batch PRs). The third countermeasure is strongly related to *trust*, both trust in the bot as well as trust in the project's own quality assurance processes. We observed that bot developers are themselves often careful with automerging. For instance, when Dependabot was acquired by GitHub in May 2019 they removed the auto-merge feature in the bot (https://github.com/dependabot/dependabot-core/issues/1973), instead urging the users to manually verify dependency updates before merging.

*"Auto-merge will not be supported in GitHub-native Dependabot for the foreseeable future. We know some of you have built great workflows that rely on auto-merge, but right now, we're concerned about auto-merge being used to quickly propagate a malicious package across the ecosystem. We recommend always verifying your dependencies before merging them."* -[Dep-1973]

Another common theme in discussions around bot adoption or discarding was *usability*. Setting up and configuring a bot is not always seen as an quick and easy task, often requiring substantial trial and error. Instead of trying to make sense of the bots manual [Rea-2673-2], many projects instead opted to set up and experiments with different settings until a satisfactory result is achieved.

*"Just tried turning on pyup.io and requires.io so we can see what they do :-)"* -[Pyt-687]

This also applied to when the bot was adopted and the contributors tried to understand what, how, and when the bot functioned. Core features that developers are particularly interested in are support for collecting everything regarding dependency updates in to one bot [Ang-19580] over having different bots for *e.g.*, different languages. Further, many feature discussions are again related to noise reduction.

*"Hmm, I hadn't heard of renovate before, but it claims to have python support and a lot of tools for reducing noise."* -[Pyt-652]

Another common usability-related challenge was that bots may not necessarily fit the workflow of the project well. We have observed both, cases where the team managed to adapt the bot as well as cases where the team changed their workflow to accommodate the bot requirements [Rea-2673-3], [Cal-16961]. We have also identified one case where a bot was outright discarded because it was judged a bad fit for the team's way of working [Gre-247].

Finally, the last usability-associated theme we identified was related to *bot promotion.* In several cases, the bot creator actively markets the bot by "popping into" relevant issue discussions in open-source software projects, nudging the project to give their bot a try. Similarly, once a project decides to adopt a bot, creators sometimes offers direct usability support by explaining or proposing ways to use the bot or helping with onboarding [Ang-20860].

Our theme analysis reveals that the key factors guiding the discussions about adoption of dependency management bots are usability, benefits and features. In turn, most of the discussion around discarding those types of bots revolved around the noise that the bot generates. Some of those factors, such as noise (*Wessel et al., 2021*) or the benefits in handling tasks at scale (*Erlenhov, de Oliveira Neto & Leitner, 2020*), have also been seen in other studies as relevant factors to, respectively, hinder or improve the development workflow.

## DISCUSSION

Central to our study is a distinction between automation tools and genuine software development bots (Devbot), as defined in *Erlenhov, de Oliveira Neto & Leitner (2020)*. We now summarise and contextualise our findings from exploring this difference based on the BIMAN dataset (*Dey et al., 2020a*). We argue that our results have multiple key implications for future research studying Devbots.

**Most automation in open-source software projects is not through (human-like) bots, but through automation scripts.** Our manual analysis of a sample of 54 widely used tools from the BIMAN dataset showed that only 10 (18.5%) comply with the Devbot definition. However, this should not be seen as criticism of the dataset, as the remaining 44 tools are certainly not false positives according to *their* definition (which classified all non-human contributors as "bots"). However, researchers need to be aware that a majority of tools contained in a dataset such as this are relatively simple automation scripts that do not exhibit any specific human-like traits, and are not qualitatively different to the kind of scripting developers have been doing for a long time. To support the study of Devbots,

new datasets (which may have to be compiled manually, or at least in a semi-automated manner) will be required.

**Dependency management is a task where Devbots are indeed common, and there are multiple widely used implementations of dependency management bots.** From the remaining 10 tools which we categorised as Devbots, nine were dependency management bots. Hence, we conclude that dependency management is the one domain where Devbots are indeed widespread and commonly used in open-source software projects. Further, multiple widely-used bots are available serving a very similar purpose. An implication for researchers of this finding is that a study of Devbots from datasets such as BIMAN is really a study of dependency management bots, as these dominate the dataset.

However, we cannot necessarily conclude from our results that dependency management bots are the only Devbots that open-source software projects use—since our study was based on a dataset of code contributions, Devbots that interact with a project in a different manner, *e.g.*, by welcoming newcomers in the issue management system (*Dominic et al., 2020*), would not emerge in our work by design. Future research will be required to assess the prevalence and impact of such other types of Devbots.

**All analysed dependency management bots exhibit similar contribution patterns.** When studying the contribution behaviour of five of these dependency management bots (Greenkeeper, Dependabot, Renovate, Pyup, and Depfu) in more detail, we observed that all five bots exhibit comparable behaviour. This indicates that these tools are indeed comparable, not only in terms of functionality but also in how they interact with developers. Consequently, the five bots identified in our research can serve as a valid starting point for future comparative studies.

We have not observed clear differences between bot commits and human commits regarding the time until PRs are resolved. This is surprising, as our results do not confirm earlier work (*Wyrich et al., 2021*), which has observed that developers handle bot contributions with lower priority than human ones. More empirical research will be required to establish if this discrepancy is due to differences in the sampling strategy, or if there are indeed certain types of bot PRs that get handled similarly fast as human contributions.

**Many open-source software projects experiment with different dependency management bots. However, sustained "co-usage" of multiple dependency management bots is rare.** A majority of 72 (77.4%) projects have used (or at least experimented with) two or more dependency management bots during their lifetime. Four projects have experimented with four of our five case study bots. This indicates that projects are not opposed to evaluating alternative bots or switching entirely. Additionally, we have observed that projects sometimes use multiple dependency bots in parallel, although this is not common outside of a "switching phase". Further research will be required to investigate reasons for the co-usage of multiple dependency management bots.

**Open-source software maintainers are hoping for improved software quality when adopting dependency management bots. Common problems when adopting these bots are usability issues, especially related to noise.** From a thematic analysis of discussions surrounding the adoption, discarding, or switching of bots we have learned

that developers predominantly expect higher code quality when using bots (*e.g.*, related to important security updates being discovered and merged earlier). Surprisingly, developers do not seem to directly expect, nor achieve, higher productivity *per se*, as adopting a dependency management bot often incurs significant noise. Particularly concerning in this context is that prominent bots such as Dependabot have even reduced their feature set related to handling noise (*i.e.*, auto-merging). This indicates that ongoing research related to the prevention of "bot spam" and bot-induced noise is timely (*Wessel & Steinmacher, 2020*), and that more research in this direction may be required. This further research will become particularly crucial if bot adoption continues to increase, as developers are currently lacking the tools to systematically deal with a large influx of bot contributions.

**Clear bot definitions are crucial to study design.** An overarching theme of our results is that, when empirically studying a somewhat "fuzzy" new concept such as bots in software engineering, great care needs to be taken to establish clear definitions of the study subject upfront. It is easy to take an existing dataset such as BIMAN because it uses the same keyword ("bot") as basis of one's own research, without realising that it may have been constructed with a different definition in mind. This bears the danger of overgeneralisation, when certain types of bots (*e.g.*, dependency management bots) are studied because they are readily available, but results are implicitly generalised to "all bots".

## Threats to validity

We now discuss the threats to the validity of our research.

**Construct validity:** Deciding on a reference framework to classify and sample bots is a challenge faced by many bot-related studies, despite the existing taxonomies in literature to support researchers (*Erlenhov, de Oliveira Neto & Leitner, 2020*; *Erlenhov et al., 2019*; *Lebeuf et al., 2019*). We mitigate this limitation by (i) using a bot taxonomy based on input from practitioners using those bots, and (ii) choosing evaluation measures or code labels (*e.g.*, PRs, issues, bot noise, trust) that have been used in previous work (*Wessel & Steinmacher, 2020*; *Wyrich et al., 2021*; *Wessel & Steinmacher, 2020*). Therefore, our findings are limited by the characteristics prevalent in such types of bots, *i.e.*, human-like traits such as communication or autonomy. In turn, starting our sample from the BIMAN dataset introduces the risk to skip bots used in that were not initially included in the dataset. Consequently, the bot activity and factors discussed in RQ2 and RQ3 are limited to our sample of projects using those bots. Future work can use our replication package to analyse a new dataset of issues and PRs mined from projects using other dependency management bots.

**Conclusion validity:** For RQ1 we quickly noticed that the GitHub projects and tool documentation often miss details that hindered our classification of bots in RQ1 using the flow-chart. Therefore, there is a risk that leads to false negatives in our sample. For instance, some tools that we did not classify as bots in our list could be bots for, *e.g.*, a Charlie user persona or an Alex persona whose bots use other team communication channels. We mitigate this threat by focusing our analysis on the distinction between true (actual bots) and false positives (tools misclassified as bots) such that the false negatives

have smaller impact on our conclusions. The limited availability of tools documentation was also a challenge in the classification done by *Dey et al. (2020b)*, hence motivating the identification based on activity patterns for the tool, instead of qualitative answers.

Moreover, comparing bot and human activity can be misleading, particularly, when evaluating time to merge PRs or close issues because the expectation on human and bot source code contributions are different. For instance, bots create many more PRs than human contributors and those bot contributions are mainly dependency updates (*Wyrich et al., 2021*). We mitigate the risk of comparing activities by delimiting our entire sample around issues with similar purpose (*e.g.*, the human created issues are inclusive of either a bot mention or comments made by dependency update bots) and by including results on bot activity per project. Moreover, one threat to our survival analysis is that KM curves are limited to detect confounding variables in data that has more than one strata (*Kaplan & Meier, 1958*). We mitigate this risk by using only one strata (bot authors) in our analysis.

**Internal validity:** During our classification for RQ1, we quickly noticed that the GitHub projects and tool documentation often miss details that would allow us to answer some of the questions in the flow-chart (*e.g.*, the first step asks whether the tool uses a chat, which is often hard to answer conclusively without using the tool). This is a limitation of the manual classification as it can lead to false negatives. For instance, some tools that we did not classify as bots in our list could be bots for, *e.g.*, a Charlie user persona or an Alex persona whose bots use other team communication channels.

In order to avoid bias during open coding for RQ3, the first and second authors had initial coding sessions until reaching agreement on a list of code labels. Then, both authors triangulated their coded labels in three different 1-h sessions twice a week until they reached theory saturation (*i.e.*, no new themes or sub-themes were found). We mitigate disagreement between coders by (i) using few and fixed labels for the PRs conversations and (ii) using definitions from literature to label the content of discussions. Examples of (ii) are the list of themes related to the benefits of using bots from *Erlenhov, de Oliveira Neto & Leitner (2020)* or the definition of noise created by bots as proposed by *Wessel et al. (2021)*. Moreover, creating distinct categories of code labels to capture the context of the PR conversation *vs* the content of the discussion allowed us to relate the discussions to the factors listed in RQ3.

**External validity:** Our findings are limited to open-source software in GitHub, since we did not collect data from other open-source software repositories or proprietary software. In other words, we analyse the projects and corresponding bot activity based on common praxis in GitHub projects, such that developers working in proprietary software may guide their discussion around new or contrasting factors to the ones listed in RQ3 such as standards defined by a company or regulatory agencies.

## CONCLUSIONS

Software engineering bots are increasingly becoming a major subject of academic study. However, despite substantial research, the question of what exactly bots are and how they differ from previously-existing automation tools still looms large. In this paper, we

contributed three-fold to this discussion. Firstly, we manually evaluated a sample of tools from an existing dataset of bot contributions, and found that only 10 of 54 tools are qualitatively different from routine automation tools. We further found that dependency management is the one domain where tools that fit our stricter definition of bots are currently in wide-spread use in open-source software projects. Secondly, we collected GitHub data for a large set of projects that use five of these dependency management bots to investigate how they are used in practice. We found that these tools have relatively similar contribution patterns, and that most projects in practice adopt different dependency management bots during their lifetime. Thirdly, we conduct a thematic analysis of discussions around bot adoption, discarding, and switching, and found that developers adopt dependency management bots to improve code quality. However, they struggle with the noise that is (sometimes) introduced by these tools.

The main implications of our study for future research are the following. Firstly, our results indicate that datasets of automated commits predominantly do not contain genuine, practitioner-perceived bot contributions. Bot researchers should take care to take this into account when analysing such data, and there may be a need for more targeted and curated datasets of bot contributions. Furthermore, researchers should consider that the practitioner-perceived bots that are contained are predominantly dependency management bots. Secondly, our results show that bot noise remains an open issue that practitioners struggle with, and which warrants further academic study.

## APPENDIX

**Table A1 IDs and corresponding URLs to the issues and comments referred in the text.**

| ID | Conversation | Theme | Issue or Comment URL |
|---|---|---|---|
| Ang-19580 | Switch | Feature | https://github.com/angular/angular-cli/pull/19580#issuecomment-743275784 |
| Ang-20860 | Adoption | Usability | https://github.com/angular/angular/issues/20860#issuecomment-364627889 |
| Cal-16961 | Adoption | Usability | https://github.com/Automattic/wp-calypso/issues/16961#issuecomment-390778832 |
| Dja-2872 | Adoption | Noise | https://github.com/pydanny/cookiecutter-django/pull/2872#issuecomment-702824915 |
| Dep-1973 | Usage | Feature | https://github.com/dependabot/dependabot-core/issues/1973#issuecomment-640918321 |
| Dyn-215 | Usage | Usability | https://github.com/Dynamoid/dynamoid/pull/215 |
| Gre-247 | Removal | Usability | https://github.com/greenkeeperio/greenkeeper/issues/247 |
| Hyp-747 | Switching | Promoting bot | https://github.com/HypothesisWorks/hypothesis/issues/747 |
| Pyt-687 | Switching | Usability | https://github.com/python-trio/trio/pull/687#issuecomment-425268701 |
| Pyt-652 | Adoption | Feature | https://github.com/python-trio/trio/issues/652#issuecomment-419605103 |
| Rea-2673-1 | Adoption | Benefits | https://github.com/react-boilerplate/react-boilerplate/issues/2673#issuecomment-501018290 |
| Rea-2673-2 | Adoption | Usability | https://github.com/react-boilerplate/react-boilerplate/issues/2673#issuecomment-501021447 |
| Rea-2673-3 | Adoption | Usability | https://github.com/react-boilerplate/react-boilerplate/issues/2673#issuecomment-500975888 |
| Str-2433 | Adoption | Usability | https://github.com/strapi/strapi/pull/2433#issuecomment-507554250 |

### Funding
This research has been funded by Chalmers University of Technology Foundation and the Swedish Research Council (VR) under grant number 2018-04127 (Developer-Targeted Performance Engineering for Immersed Release and Software Engineers). The funders had no role in study design, data collection and analysis, decision to publish, or preparation of the manuscript.

### Grant Disclosures
The following grant information was disclosed by the authors:
Chalmers University of Technology Foundation and the Swedish Research Council (VR): 2018-04127.

### Competing Interests
Philipp Leitner is an Academic Editor for PeerJ CS.

### Author Contributions
- Linda Erlenhov conceived and designed the experiments, performed the experiments, analyzed the data, performed the computation work, prepared figures and/or tables, authored or reviewed drafts of the paper, and approved the final draft.
- Francisco Gomes de Oliveira Neto conceived and designed the experiments, performed the experiments, analyzed the data, performed the computation work, prepared figures and/or tables, authored or reviewed drafts of the paper, and approved the final draft.
- Philipp Leitner conceived and designed the experiments, performed the experiments, analyzed the data, performed the computation work, authored or reviewed drafts of the paper, and approved the final draft.

### Data Availability
A replication package is available at Zenodo: Linda Erlenhov, Francisco Gomes de Oliveira Neto, & Philipp Leitner. (2021). Dependency Management Bots in Open-Source Systems-Prevalence and Adoption [Data set]. Zenodo. https://doi.org/10.5281/zenodo.4974219.

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
