# Peer review of "Dependency management bots in open-source systems—prevalence and adoption"

_PeerJ Computer Science, doi:10.7717/peerj-cs.849_

## Round 0.1 · original submission · Major Revisions

Dear Authors,

To begin with, I want to compliment you on this monumental task that you are undertaking in this interesting and relevant field. There are some really interesting data in the article and although in some cases published in earlier work, it is good to combine these into one journal article with extra findings and analysis added. Furthermore, the work is easy to read and well written. We are therefore willing to move forward with this work, but under the condition that the comments from the reviewers are taken very seriously.

The reviewers are relatively strict about several things and as editor I support their views and comments. Furthermore, I add some of my own comments to theirs:

1. It is common to also create a reproducibility package with the article.
2. Please indicate explicitly, perhaps with an explaining paragraph in the introduction or at the end, what is new in this article.
3. Please further explain the qualitative discussion, as per comments from R2.
4. Please improve the related work and citations.
5. Please align all numbers to the right, to make them easier to compare.
6. "the risk it"
7. Instead of the many footnotes, would it be better to create a data table in an appendix at the end of the article with all issues cited? Could be interesting to classify them further as well, as they are in a way rough data?
8. Under external validity, don't you wish to claim smth about generalizability towards non-GH projects?
9. Avoid the word "important".
10. Under discussion I would have expected some more high-level insights about the use of bots in SE in general. Atm, the work is a little shallow and factual, whereas it would be great to get a better understanding of what your work means for the theoretical foundation under empirical SE and empirical SE analysis. This is where the work is really lacking and taking more of a helicopter view will help you not only in this work, but in future endeavours as well.

Please ensure to create a reply letter indicated the changes you have made. We look forward to working with you on this further.

With kind regards,

Your Associate Editor

Reviewer 1 ·

Basic reporting

This paper presents a mixed-methods study on the impacts of adopting a specific type of bot: the dependency management bots. The study examines an important and timely topic. Bots as software maintenance task automation tools have become increasingly popular, as discussed in the paper. The current state-of-art is clearly and accurately summarized. The paper is very well written, easy to read and understand. In addition, the problem is well-motivated. Furthermore, the paper outlines the main takeaways and implications for the open-source community and bot researchers.

Regarding the replication package and supplemental materials, I would also recommend sharing a codebook with code names, descriptions, and examples of quotations for all themes and conversation labels. It would facilitate the understandability of each one of them.

Experimental design

Regarding the experimental design, there are a few places to clarify or improve.

First, it would make the paper stronger if the authors could clarify the motivation of focusing on dependency management bots after conducting a broader study that attempted to categorize several "bots" and assess how this classification would provide meaningful insights about bots adoption and its impacts. Even though all encountered bots are dependency management bots, It still seems the paper has two disjunct parts. The first part focused on classifying the bots from the BIMAN dataset using the previous categorization by Erlenhov et al, 2020. The second one restricted the analysis to a specific type of bots: dependency management bots. I believe the findings reported in the RQ1, which resulted in the decision of focusing on dependency management bots, are closely related to the type of bots/automation present in the BIMAN dataset. Those bots/automation are the ones that commit code, which is also the case for dependency management bots. It is not expected, for example, to see any bot that only automates tasks and post comments on issues.

Another opportunity to strengthen the paper is to clarify the qualitative analysis method in Section 6.2. Overall, it is not clear how the authors conducted the coding process from the description. Have multiple authors participated in the coding process? Have they discussed the codes? Have they reached an agreement? The authors should enhance the analysis method in the revision if it was not rigorous and describe the method clearly.

In the revision, I also encourage the authors to articulate in the discussion *how* this study complements the findings from previous works: Which prior findings are confirmed? Which prior findings are challenged or extended? What findings are new? How are the new findings fit (or not fit) into the previous works?

Validity of the findings

Overall, the paper is clean, readable, addresses a significant problem, presents valid results grounded on the data provided. I do feel the paper has the weaknesses above that should be fixed before publication. I recommend major revisions, as I consider all of the comments could be addressed over the revision period.

Additional comments

Additional (minor) comments:
- Line 28: “creating pull requests (PRs) (Wessel et al., 2020a) [...] or code contributions over time (Wessel et al., 2018).” -> double-check the references and their examples.

- Line 30: The last sentence of the introduction first paragraph lacks a reference.

- The link in the first quotation of Section 6.3 (https://github.com/pydanny/ cookiecutter-django/pull/2872#issuecomment-702824915) is redirecting to the wrong URL (https://github.com/pydanny/). The same occurred with the link in the third quotation.

- Line 490: “[...] have been used in previous work (Wessel and Steinmacher, 2020; Wyrich et al., 491 2021; Wessel and Steinmacher, 2020).” -> Wessel and Steinmacher, 2020 were cited twice.

Reviewer 2 ·

Basic reporting

*The English language should be improved to ensure that an international audience can clearly understand your text.
Some examples where the language could be improved include sections 3, 5.1, 6.1, 7.


*The problem definition has not been formulated clearly. I think the motivations for this study need to be made clearer. The authors must do a better job to explain why their work is better than what came before or after.

*This paper is not self-contained. In almost all sections, the authors refer to their previous publications. I think this paper suffers from extensive self-plagiarism. Additionally, the main contribution and innovation of this paper are not clear enough.

* The audience of this paper cannot understand it without reading the publications from these authors.


*The citation format sometimes isn't consistent. For instance, Erlenhov et al. (2020) and (Erlenhov et al., 2020).

* In section 4.2, there is three self-citation to the same paper in one paragraph.


The authors refer to this paper as one of the references in the study methodology section and 6.1 Data Collection (Erlenhov et al., 2021).

Experimental design

*In Figure 2, the flowchart is not clear enough. A better explanation is required to indicate the sources of knowledge for designing such a decision model. Would you please add your evidence for this figure?


*The author mentioned the BIMAN approach, but they should elaborate on this dataset further. Currently, there is no explanation has given.


* The authors mentioned several taxonomies in the related work section, and they introduced their paper as one of the papers in related work. I think that the paper needs more effort and citations from other studies to explain the taxonomies.

Validity of the findings

* How did the study address these questions?
and it needs more detail about research methods.

---

## Round 0.2 · accepted · Accept

The reviewers have achieved consensus about the fact that this work is now of acceptable quality. I agree with them about the timeliness and the quality of the work. Furthermore, I would like to add my compliments for the processing of the comments: both reviewers have little further comments.

I congratulate you with the publication of this work and am content about the fast turnaround for this timely article.

Reviewer 1 ·

Basic reporting

Thank you to the authors for providing this detailed revision. I would reinforce that I find the topic is very important and timely. I also think the study is well-executed, and the paper is well-written, easy to read and understand. In addition, the problem is well-motivated.

Due to the detailed revision version provided by the authors, I recommend paper acceptance. The authors addressed all the priority and significant review points I have raised. I see no major issues related to the approach followed in conducting the study or even the results' presentation.

Experimental design

There are no further suggestions for improvement.

Validity of the findings

Overall, the paper is clean, readable, addresses a significant problem, presents valid results grounded on the data provided.

Reviewer 2 ·

Basic reporting

-

Experimental design

-

Validity of the findings

-

Additional comments

The authors have significantly improved the manuscript by restructuring and condensing the previous draft.